# Exploring the Mediating Effects of Cognitive Function, Social Support, Activities of Daily Living and Depression in the Relationship between Age and Frailty among Community-Dwelling Elderly

**DOI:** 10.3390/ijerph182312543

**Published:** 2021-11-28

**Authors:** Lin-Yen Chen, Tzu-Jung Fang, Yu-Chih Lin, Hsiu-Fen Hsieh

**Affiliations:** 1Department of Nursing, Kaohsiung Medical University Hospital, Kaohsiung Medical University, Kaohsiung 807, Taiwan; e0955058335e@gmail.com; 2Division of Geriatrics and Gerontology, Department of Internal Medicine, Kaohsiung Medical University Hospital, Kaohsiung 807, Taiwan; tzujung66@gmail.com; 3Division of General Internal Medicine, Department of Internal Medicine, Kaohsiung Medical University Hospital, Medical University, Kaohsiung 807, Taiwan; springfred@gmail.com; 4Department of Medical Humanities and Education, School of Medicine, Kaohsiung Medical University, Kaohsiung 807, Taiwan; 5School of Nursing, College of Nursing, Kaohsiung Medical University, No.100, Shih-Chuan 1st Road, Kaohsiung 807, Taiwan; 6Department of Medical Research, Kaohsiung Medical University Hospital, Kaohsiung Medical University, Kaohsiung 807, Taiwan

**Keywords:** older adults, frailty, activities of daily living, cognitive function, depression

## Abstract

With 16.15% of its total population aged 65 or above, Taiwan is already an aging society. Frailty is a natural consequence of aging, which may decrease physical strength and deteriorate physiological functioning. We examined the mediating effects of cognitive function, social support, activities of daily living (ADL), and depression in the relationship between age and frailty in older people living in the community. This cross-sectional study used a structured questionnaire to collect data from a convenience sample of 200 pre-frail to mildly frail older adults in southern Taiwan. Structural equation modeling was used for data analysis, with data collected from July to November 2020. ADL mediated the relationship between age and frailty, while cognitive function also mediated the relationship between age and frailty, indicating that ADL and cognitive function were significant determinants of frailty. The path from age to frailty was significant, indicating that age was a significant determinant of frailty. The standardized total effect of age affected frailty through the mediating roles of ADL and cognitive function. Age, depression, ADL, and cognitive function explained 59% of the variance in frailty among older adults. ADL and cognitive function are significant mediators of frailty among older adults.

## 1. Introduction

According to the World Health Organization (2002), when 14% of a nation’s population is over 65 years of age, it is termed an “aged society” [1]. There are 3.8 million (16.15%) residents aged 65 or older currently in Taiwan (Ministry of Health and Welfare, 2021). In particular, the experience of frailty has been an important and emerging issue for healthcare providers in recent years. Taiwan’s government has been actively promoting advanced long-term care for their aging society, to slow down the process of frailty among community-dwelling older adults for a better quality of life [2].

Frailty is identified by the presence of unintentional weight loss, exhaustion, muscle weakness, low levels of ADL, and slow gait [3], and is characterized by diminished muscle strength, endurance, and reduced physiologic function that increases vulnerability for dependency and event death [4]. In the United Kingdom, 7% of the elderly were categorized as frail, 47% were pre-frail, and 46% were not frail [5]. In the United States, 7% of the 65-and-older and 25–80% of the 80-and-above demographic populations are frail [6]. However, in Taiwan, the Ministry of Health and Welfare (2019) reported that more than 50% of people aged 65 and older are in the pre-frail stage, 11.2% are frail, and those of a more advanced age experience more severe frailty.

According to the stereotype embodiment theory developed by Levy (2009), the lifetime exposure to cultural messages of ageism leads to an internalization of ageist constructs. In this study, we used the stereotype embodiment theory, and focus on the three tracks of this theory, including psychological, behavioral, and physiological. Aging probably is negatively associated with cognitive function and social support, and is positively associated with depression; all these age-related factors can directly and/or indirectly influence their physical health (frailty) [7].

Older adults with cognitive impairment stood out, with 12- to 13-fold increased prevalence and incidence of disability [8]. Sternberg et al. emphasized that disability, cognition, and mood are important elements for developing frailty [9]. Social support was significantly and negatively associated with frailty [10,11], and it was indicated social support interventions have a substantial effect on frailty among the elderly [10].

It was reported that frailty, exhaustion, slowness, inactivity, age, and weakness are closely related to cognitive deterioration among older adults living in the community [12]. One study suggests that cognitive function and frailty interact within a cycle of decline associated with ageing [13].

Therefore, we hypothesized the following:

**Hypothesis** **1** **(H1).**
*Age is associated with frailty, cognitive function, and ADL.*


**Hypothesis** **2** **(H2).**
*Cognitive function has a significant negative association with frailty.*


**Hypothesis** **3** **(H3).**
*Cognitive function has a significant positive association with ADL.*


**Hypothesis** **4** **(H4).**
*Social support has a significant and negative association with cognitive function.*


**Hypothesis** **5** **(H5).**
*Social support has a significant negative associated with frailty.*


**Hypothesis** **6** **(H6).**
*Cognitive function plays a role as a mediator in the relationship between age and frailty.*


Insufficient ADL are related to higher probability for frailty for older adults [14,15]. Another study also illustrated that ADL might predict pre-frailty and frailty in older adults [16].

Therefore, we hypothesized the following:

**Hypothesis** **7** **(H7).**
*ADL have a significant and negative association with frailty.*


**Hypothesis** **8** **(H8).**
*ADL play a role as a mediator in the relationship between age and frailty.*


A systematic literature review found that older adults with depression are more likely to develop frailty than those without depression [17]. People with depression have a loss of interest in activities (WHO, 2021), while having a lower motivation to participate in physical or social activities, making it difficult to build and maintain interpersonal relationships. Therefore, depression can severely impact the ADL of older adults [18]. Depression can reduce social activities, and this results in impairment of memory, concentration, and a decrease of ADL, which in turn affects cognitive function [19]. Figure 1 presents the conceptual model developed and tested in this study, which was based on the stereotype embodiment theory (SET) developed by Levy (2009).

Following this, we hypothesized that:

**Hypothesis** **9** **(H9).**
*Depression has a significant positive association with frailty.*


**Hypothesis** **10** **(H10).**
*Social support has a significant negative association with depression.*


**Hypothesis** **11** **(H11).**
*Depression has a significant positive association with cognitive function.*


**Hypothesis** **12** **(H12).**
*Depression has a significant negative association with ADL.*


## 2. Materials and Methods

### 2.1. Participants and Setting

This study adopted a cross-sectional survey design. Participants were recruited from a medical center in Taiwan through convenience sampling. The inclusion criteria were (1) being aged 65 or above, and (2) having a Clinical Frailty Scale (CFS) score of 4–6. The exclusion criteria were (1) having vision impairment or vision loss, (2) existing mental disorders, and (3) a CFS score < 7.

According to MacCallum et al. [20], in a priori sample size calculation, the minimum sample size for our model structure would be 200. Hence, in this study, a total of 200 frail older people were recruited.

### 2.2. Instruments

Demographic characteristics were recorded, including gender, age, education level, marital status, living status, activity status, main caregiver, body mass index, chronic disease history, exercise pattern, current source of monthly income, status as a volunteer, and being on social welfare.

#### 2.2.1. Clinical Frailty Scale

The CFS is a nine-point scale summarizing the overall level of fitness or frailty of an older adult after they have been evaluated by a health care professional [21]. The CFS is a judgment-based tool, scored so that higher scores indicate greater risk of developing frailty. The CFS measures how adults have moved, functioned, thought, and felt about their health over the previous 2 weeks [22]. It is divided into five levels, with scores of 1–3 points classified as healthy; 4 as fragile; 5–7 as mild, moderate, or severe frailty; 8 as very severe vital signs; and 9 as the elderly who are dying. It is currently commonly used in geriatrics, and each ascending level entails an increased risk of death or hospitalization by 21.2% and 23.9%, respectively. Its reliability and validity are also strong. The Cronbach’s alpha was 0.88 for the overall Chinese version [23].

#### 2.2.2. Short-Form Geriatric Depression Scale

The 15 items that constitute the Geriatric Depression Scale Short Form (GDS-SF) were extracted from the 30-item version [24]. There is a high correlation between the GDS full and short forms (r = 0.91, *p* < 0.01). The GDS-SF score range from 0–15: 0–4 indicates no depression, 5–9 indicates mild depression, and 10–15 indicates moderate to severe depression. Analyses were conducted with both the GDS and GDS-SF. The GDS-SF’s Cronbach’s alpha was 0.72. It has been widely used to screen for depression among older adults both in the community and in acute medical or long-term care institutions [23].

#### 2.2.3. Short Portable Mental Status Questionnaire

The Short Portable Mental Status Questionnaire (SPMSQ) is used to assess organic brain deficits in older patients [25]. The SPMSQ score is derived from the number of errors based on a 10-item list, coding errors as “1” and correct answers as “0”. The SPMSQ scale content covers time, place, sense of orientation, memory, current affairs, calculations, among others. Items include tasks on orientation (e.g., “What is the date today?”), memory (e.g., “What is your mother’s maiden name?”), and attention (e.g., “Subtract 3 from 20 and keep subtracting 3 from each new number, all the way down”). Thus, individual cognitive scores range from 0 to 10, with a lower value indicating better cognitive performance. A cutoff score of 3 is generally used. The patient’s education level may influence the test result. Depending on the score, four classifications of disorders are attributed: scores of 0–2 indicate normal functioning, scores of 3–4 indicate the presence of cognitive deterioration, scores of 5–7 indicate moderate cognitive impairment, and scores of 8–10 indicate severe cognitive impairment. The higher the SPMSQ score, the worse the cognitive function Cronbach’s alpha was 0.82 [26].

#### 2.2.4. Activity of Daily Living

The Barthel Index is used to assess activity of daily living in older adults, and consists of 10 items assessing the ability to achieve certain activities without requiring assistance. The activities are feeding, moving from a wheelchair to a bed and vice versa, personal toileting, getting on and off the toilet, bathing oneself, walking on a level surface, ascending and descending stairs, dressing, and both bowel and bladder control. The score ranges from 0–100, with scores from 0–20 indicating complete dependence, 21–60 indicating severe dependence, 61–90 indicating moderate dependence, 91–99 indicating mild dependence, and 100 indicating complete independence. The index’s inter-rater agreement is 0.90, and Cronbach’s α value is between 0.92 and 0.93 [27]. Currently, this scale is most commonly used to assess physical function in long-term care patients in Taiwan.

#### 2.2.5. The Social Support Scale

The Social Support Scale was specially designed by the Institute of Social Support for the Elderly. It has 12 simple and clear questions, divided into three types: affective, self-esteem, and instrumental support. A 5-point Likert scale is used for answering, to collect, measure, and score quantitative data. The options are “strongly disagree”, “disagree”, “no opinion”, “agree”, and “strongly agree”, giving each item 1, 2, 3, 4, or 5 points respectively. The higher the score, the higher the degree of the variable measured. The overall Cronbach’s α value of the scale is 0.89, and the α coefficient value of each dimension ranges from 0.71 to 0.85, indicating strong reliability and suitability for use with older participants. The scale was used with the permission from the original author [20].

### 2.3. Data Collection and Ethical Considerations

Older people with significant frailty were selected using the convenience sampling method. After receiving approval from the relevant Institutional Review Board (KMUHIRB-E(II)-20200158), the principal investigator (PI) visited outpatient departments housing older adults and their families, and explained the purpose and processes of the study to all eligible participants. Data were collected from July to November 2020. Upon obtaining informed consent, each participant received a package including an informed consent form, the questionnaires, and an envelope. The participants completed the questionnaire at their convenience, and called the PI to collect it afterward. Each participant completed hard copies of the demographic and structured questionnaires on the same day they gave consent to participate. The data collection procedure took 40–60 min to complete.

### 2.4. Data Analysis

All data were analyzed using SPSS version 21.0 for Windows (IBM Corp., Armonk, NY, USA) and AMOS version 21.0 (SPSS Inc., Chicago, IL, USA). The CFS, Barthel Index, SPMSQ, and GDS-SF data were subjected to descriptive statistics, including analyses of frequency, percentage, range, mean, and standard deviation. Bivariate correlations were used to examine the relationships between variables. For data analysis and hypothesis testing, a confirmatory factor analysis of the initial measurement model and the maximum likelihood method were used for data fitting. Structural equation modeling (SEM) was used to test model fit. The model fit was examined using chi-square/df (χ2/df), goodness of fit index (GFI), adjusted GFI (AGFI), and the root mean square error of approximation (RMSEA). A good model fit is indicated by a non-significant χ2 /df value, GFI and AGFI greater than 0.90, and an RMSEA of less than 0.05. The Sobel test is a method of testing the significance of a mediation effect. The Sobel test is a z test, in which a z value ≥1.96 suggests that meditation may exist [28].

## 3. Results

### 3.1. Participant Characteristics

A total of 200 participants exhibited clinical frailty scores from 4 to 6. Specifically, about 48.5% of participants scored 4, and 51.5% scored 5 or 6, with participant characteristics shown in the Table 1. In addition, the mean score of the Barthel Index was 84.3 (SD ± 18.2), with 80 participants (40%) being moderately dependent. The SPMSQ mean score was 2.0 (SD ± 2.5). The GDS-SF mean score was 3.1 (SD ± 2.6). As for the Social Support Scale, the mean score was 48.6 (SD ± 14.2).

### 3.2. Preliminary Analysis

The Pearson’s correlation related factors used to analyze the degree of frailty and age (r = 0.55, *p* < 0.001), cognitive dysfunction (r = 0.49, *p* < 0.001), and the depression (r = 0.39, *p* < 0.001) showed a significant positive correlation. While the degree of frailty was correlated with the level of education (r = −0.27, *p* < 0.001), and ADL (r = −0.69, *p* < 0.001), these were significantly negatively correlated. In addition, there was no statistically significant relationship between frailty and social support (r = −0.103, *p* = 0.80) (Table 2).

### 3.3. Structural Equation Model

Statistics of the model showed a good fit: χ2/df =1.337, degree of freedom = 81, *p* = 0.023, GFI = 0.933, AGFI = 0.900, CFI = 0.975, NNFI = 0.967, IFI = 0.975, and RMSEA = 0.041. The structural relationships with the standardized coefficients are presented in Figure 2. The Sobel test was used to examine the indirect effects via SEM.

This showed a mediating effect of ADL on the relationship between age and frailty (z = 3.636, *p* < 0.001), and another mediating effect of cognitive function on the relationship between age and frailty (z = 2.887, *p* < 0.001), indicating that older adults’ ADL and cognitive function were significant determinants of frailty. The path from age to frailty was significant (z = 5.333, *p* < 0.001), indicating that age was a significant determinant of frailty. The standardized total effect of age affected frailty through the mediating roles of ADL and cognitive function (z = 9.167, *p* < 0.001) (Table 3). In the final model, age, depression, ADL, and cognitive function explained 59% of the variance in older adults’ frailty (Figure 2). 

## 4. Discussion

Under the SET, among older adults with frailty, age was positively correlated with frailty but negatively correlated with ADL and cognitive function (supported by H1). Cognitive function and ADL were significantly and negatively associated with frailty (supported by H2 and H7). Cognitive function has a significant positive association with ADL (supported by H3). Depression has a significant negative association with social support, cognitive function, and ADL (supported by H10-H12). In this model of parallel multiple mediators for frailty, ADL and cognitive function played mediating roles, and weakened the effects of age (supported by H6 and H8).

Our results were compatible with previous findings from many studies, where the process of aging is a risk factor for frailty, cognition, and psychosocial function [29]. In addition, we found that the effect of age on frailty was partially mediated by ADL, and this result corresponded to Hao’s study where poor ADL partially and indirectly leads to frailty [30]. This result was also supported by another study showing exercise interventions could alleviate frailty in older adults [31]. In other words, the degree of frailty can be alleviated by increasing ADL, and this indicated that the progression of frailty in older adults can be slowed down by maintaining regular ADL [16,32]. Clinical Practice Guidelines have been developed for the management of frailty among older adults, with a recommendation to maintain with a resistance training component [33].

The effect of age on frailty was also partially mediated by cognitive function in our study. This result was supported by a previous study that impaired cognitive function led to frailty [9]. It has been shown that both age and cognitive function decline among older adults contributed to frailty [34], and cognitive function was significantly related to frailty [35]. ADL was found to be negatively correlated to cognitive function in our study, and lower ADL is associated with poorer cognitive function, leading to greater frailty [36]. Therefore, cognitive decline can affect frailty among older adults.

One possible cause of cognitive deterioration is patients with degenerative brains or their family members may often mistake this as forgetfulness. Family members should seek medical attention when their older relative’s condition is severe and apparent by poorer physical functioning and daily life activity; however, it is possible that the cognitive function deterioration is significant with severe frailty at this moment, so it is recommended that older adults should be screened for early signs of cognitive deterioration, as many interventions can be provided to slow down cognitive deterioration and subsequent frailty.

In addition, it was found that social support had no significant association with frailty and cognitive function (not supporting H4 and H5), and our result was similar to another study where there were no associations between frailty and the social support variables, except for housekeeping [37]. A possible reason for the insignificant impact of social support on frailty in this study is that approximately 91% of participants lived with their families, and they might be influenced by the traditional Chinese culture of Confucianism [38], which emphasizes possessing an internal moral compass or conscience where older Chinese adults take support for granted.

In other studies, different from our results, social support was regarded as an element against frailty [39]. The International Conference of Frailty and Sarcopenia Research (ICFSR) also developed clinical practice guidelines for the identification and management of frailty in older adults. It strongly recommended that all persons with frailty should receive social support to improve or slow down the process of their frailty [39]. One possible reason for the difference is that our study only recruited older adults who had a CFS score ≥ 7 (mild to moderate frailty), without results showing that depression did not have a significant direct effect on frailty but an indirect effect mediated through its effect on activities of daily life (ADL), where less social and physical activities were features of older adults with depression [40] (not supported by H9), and this result is different from previous research [40] showing that individuals with depression were more prone to develop frailty. In addition, we found that depression was negatively associated with social supports and ADL, and this result is similar to previous studies [41], including Hsu & Yeh. [42]. Our results suggested that depression might have indirect effect on developing frailty.

Under traditional Chinese culture, with more concerns of “face”, “endurance” and “stigma”, older Chinese adults with depression demonstrate more somatic complaints [43] for their emotions, so most older adults with depression might first seek treatment in a general medical setting instead of professional psychiatric assistance, and where poorer interpersonal relationships reveal reduced social support [41]. Some studies have also shown that a lack of social support is associated with increased frailty among older adults [44,45].

In addition, we also found that depression has a significant positive association with cognitive impairment. Depressive symptoms might lead to lower cognitive capacity, including the speed of thinking processes, visual selective attention, working memory and executive functioning [46], so early effective interventions on people with depression could alleviate their depression as well as cognitive impairment [47].

It is accordingly suggested that when older adults frequently complain of physical discomfort, an assessment should be conducted to detect symptoms of depression, and furthermore, good interpersonal relationships with strong social support should be established for improving depressive symptoms [48], with subsequent reduction of frailty.

In sum, it is recommended that older adults should maintain regular daily activities, exercise, and have more social interaction to improve their cognitive function as well as slow down the processes of frailty.

## 5. Limitations

Our study had some limitations. Firstly, this is a cross-sectional study, and only older adults with mild-to-moderate frailty were enrolled, so future studies with long-term follow-up are needed to investigate all older adults with varying degrees of frailty to have a deeper understanding of the associations among cognitive function, social support, ADL, depression, and frailty. Secondly, the “self-perception of being old” of older adults was not measured, and this should be included in future studies. Thirdly, only older adults who had a CFS score < 7 (mild to moderate frailty) were recruited, possibly leading to social support not correlating with cognitive function or frailty. Future studies of those aged 65 and over living with severe frailty should be included. In this study, the social support scale may not have measured our participants’ needs. In future, researchers need to select questionnaires of social support for measuring older adults with frailty.

## 6. Conclusions

This study investigated the relationships between frailty, cognitive function, and ADL among older adults, with the findings demonstrating a mechanism from aging to frailty based on the SET. The parallel multiple mediator models in this study could potentially contribute to the prevention and intervention of frailty in older adults. The completed mediation effects of cognitive function and ADL were confirmed. Depression did not have a significant direct effect on frailty, but an indirect effect that was mediated through its effect on activities of daily life (ADL).

## Figures and Tables

**Figure 1 ijerph-18-12543-f001:**
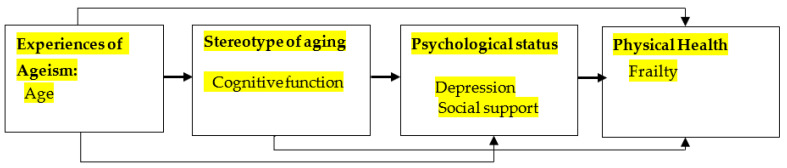
Conceptual framework of hypothesized model.

**Figure 2 ijerph-18-12543-f002:**
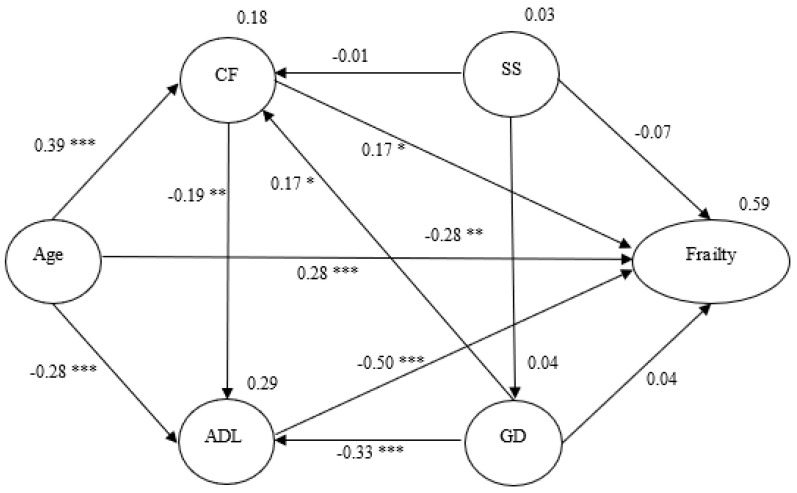
Hypothesized model with standardized estimates. Note. Social support- SS; Cognitive function- CF; Activities of daily living- ADL; Geriatric depression- GD; →-unidirectional path; * *p* < 0.05, ** *p* < 0.01, *** *p* < 0.001.

**Table 1 ijerph-18-12543-t001:** Descriptive statistical results of basic demographic data (*n* = 200).

Item	Mean	SD	*n*	%
Age	78.9	7.4		
Height	156.5	8.6		
Body weight	62.3	11.9		
Body mass index	25.4	4.2		
Number of chronic diseases	4.1	1.2		
Religious beliefs				
No			37	18.5
Yes			163	81.5
Activity status				
Free movement			144	72
Assistive devices are required for the activity, but no assistance from others is needed			45	22.5
Activities require assistance from others			11	5.5
Serves as a volunteer				
No			185	92.5
Yes			15	7.5
Uses social welfare resources				
No			179	89.5
Yes			21	10.5
Gender				
Male			83	41.5
Female			117	58.5
Education level				
Illiterate			43	21.5
Elementary school			70	35
Junior high school			15	7.5
Senior high school			32	16
Junior college			17	8.5
Above college			23	11.5
Marital status				
Unmarried			2	1
Married			142	71
Widowed			54	27
Cohabitation (partner)			2	1
Exercise time/week				
≤1 h			59	29.5
1–3 h			53	26.5
3–5 h			37	18.5
5–8 h			38	19.0
>8 h			13	6.5
Monthly income				
US$ < 107			43	21.5
US$ 107–178			92	46
US$ 179–356			17	8.5
US$ 357–535			13	6.5
US$ ≥ 536–714			35	17.5

**Table 2 ijerph-18-12543-t002:** Degree of frailty related to age, education level, activities of daily living, cognitive function, depression, and social support (*n* = 200).

	1.	2.	3.	4.	5.	6.	7.
1. Age	1						
2. Education	−0.300 **	1					
3. ADL	−0.388 **	0.213 **	1				
4. CF	0.404 **	−0.438 **	−0.369 **	1			
5. GD	0.210 **	−0.219 **	−0.372 **	0.263 **	1		
6. SS	−0.032	0.144 *	−0.004	−0.102	−0.315 **	1	
7. Frailty	0.547 **	−0.273 **	−0.687 **	0.486 **	0.395 **	−0.103	1

Note. Social support—SS, Cognitive function—CF, Activities of daily living—ADL, Geriatric depression—GD, * *p* < 0.05, ** *p* < 0.01.

**Table 3 ijerph-18-12543-t003:** Parallel multiple mediator model for effect of age on frailty.

Relationship	Point Estimate			Bootstrapping				
		Product of Coefficients		Percentile 95% CI		BC 95% CI		
		SE	Z	Lower	Upper	Lower	Upper	*p*
	Indirect Effects							
Age to ADL to F	0.006	0.009	2.302	0.000	0.028	0.000	0.027	0.041
Age to CF to F	0.008	0.003	2.280	0.003	0.014	0.003	0.016	0.003
	Direct Effects							
Age to F direct	0.046	0.010	6.571	0.023	0.058	0.027	0.062	0.001
	Total Effects							
Age to F total	0.060	0.006	10	0.048	0.070	0.047	0.069	0.001

Note. Activities of daily living—ADL, Cognitive function—CF, Frailty—F, SE—Standard error, 2000 bootstrap samples, Chi-square = 108.280, Degree of freedom = 81, *p* value = 0.023, Chi square/df =1.337, GFI = 0.933 AGFI = 0.900 CFI = 0.975, NNFI = 0.967 IFI = 0.975, RMSEA = 0.041.

## Data Availability

Not applicable.

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
