# Peer review of "Exploring the Mediating Effects of Cognitive Function, Social Support, Activities of Daily Living and Depression in the Relationship between Age and Frailty among Community-Dwelling Elderly"

_ijerph, 2021, doi:10.3390/ijerph182312543_

Round 1

Reviewer 1 Report

This is an interesting article, largely well written except for some problems with English grammar and syntax.

My comments and suggestions are mostly on minor details

Lines:

43: Full stop after 'years':   ...years. The Taiwan...

71: Change to: 'Cognitive function is significantly and negatively associated with frailty '   or even better:  'Cognitive function has a significant negative association with frailty'

77 and 84: Similar changes as above.

80:  Insert 'to'.   ..related to higher...  

113: Delete 'using'

117: change to [20] instead of(1996).  Delete [20] in line 119

127: include reference to CFS scale. '...professional [ref].

127: insert 'is':  '..CSF is a ...

210: change 'was showed' to: 'are shown'

Table 1: 'Number of chronic disease'  Changes to 'Number of chronic diseases'

 Actitity status:  Delete symbols in n and % columns

Use social welfare resources adults:  Delete 'adults'

Education level, n (%):  Delete ', n (%)'

Widow: Change to 'Widowed'     (to include both widows and widowers) 

Exercise type/week: Change to 'Exercise time/week'

1 hour/week: Delete '/week', change to ' ≤ 1 hour'  and similar for next 4 lines

Monthly income: Explain NTD in subscript: 1 NTD=approx .... US, UK or Euro 

Table 2. Include in the table heading that it also includes bivariate correlations between other items.

Make clear whether physical activity (PA), activity of daily life (ADL) and exercise (Table 1) refer to the same item or not.  It is confusing that Table 1 refers to both activity status and exercise, Table 2 refers to physical activity in the table heading and to ADL in the table, Table 3 refers to PA in the table and to ADL in the note explaining the acronyms in the table. Use the same name consistently for what is the same thing.

260: physical activity = ADL or hours of weekly exercise?

260: 'Significantly negative' : Change to 'significantly and negatively'

261: postive > positive

262: significantly > significant

274: 'were developed' > have been developed

275: 'and it recommended' > 'with a recommendation'

279: delete 'of'

284: suggest: 'One possible cause of cognitive deterioration is degenerative brain disease, and patients.......'

289: 'detecting potentially': change to 'signs of'

293: 'supported by' Change to 'supporting'

300:'Without significant impact on frailty in our study' :  suggest change to 'In other studies, different from our results, social support......[38]'

306: '....did not have a significant effect on frailty...'  Consider the following: ' ..did not have a significant direct effect on frailty, but an indirect effect mediated through its effect on activities of daily life (ADL)'

310-311: This statement would then be covered by what is suggested above.

315: insert a comma after  'assistance'

315: 'a reduction a' : change to 'reduced'

316: Delete 'people with'

317: 'port increase frailty' change to :'port is associated with increased frailty'

318: 'found depression': insert 'that': 'found that depression' 

320: 'cludes': change to 'cluding'

Author Response

43: Full stop after 'years':   ...years. The Taiwan...

A: As recommendation, we have revised “,” to “.” And “the” to “The” Taiwan…’’ (Line 45).

71: Change to: 'Cognitive function is significantly and negatively associated with frailty '   or even better:  'Cognitive function has a significant negative association with frailty'

A: As recommendation, we have revised “cognitive function was a significantly negative associated with frailty.” to “cognitive function has a significant negative association with frailty” (Line 77).

77 and 84: Similar changes as above.

A: We have revised some sentences in the manuscript (Lines 81 and 91).

80:  Insert 'to'.   ..related to higher...

A: We have revised some sentences in the manuscript (Lines 81 and 91).

117: change to [20] instead of (1996).  Delete [20] in Line 119

A: 

We have changed to [19] instead of (1996) and deleted [19] in lines134 and 136.

Because references 9 and 11 are duplicated, the reference [20] is changed to [19].

127: include reference to CFS scale. '...professional [ref].

A: We have inserted the reference of CFS scale in lines 144 and 462.

127: insert 'is':  '..CSF is a ...

A: As recommendation, we have insert “is” on line144.

210: change 'was showed' to: 'are shown'

A: As recommendation, we have changed “was showed” to “are shown” (Line 228).

  1. Table 1: 'Number of chronic disease' Changes to 'Number of chronic diseases'

 Actitity status:  Delete symbols in n and % columns

Use social welfare resources adults:  Delete 'adults'

Education level, n (%):  Delete ', n (%)'

Widow: Change to 'Widowed'     (to include both widows and widowers) 

Exercise type/week: Change to 'Exercise time/week'

≤ 1 hour/week: Delete '/week', change to ' ≤ 1 hour'  and similar for next 4 Lines

Monthly income: Explain NTD in subscript: 1 NTD=approx .... US, UK or Euro 

A: As recommendation, we have changed “Number of chronic disease” to “Number of chronic “diseases”, “Widow” to “Widowed”, “Exercise type/week” to “Exercise time/week” and revised Monthly income “NTD” to “US$” (Table 1).

 In addition, we have deleted “n (%)”, “adults” and “/week”, etc. (Table 1).

Table 2. Include in the table heading that it also includes bivariate correlations between other items.

Make clear whether physical activity (PA), activity of daily life (ADL) and exercise (Table 1) refer to the same item or not. 

It is confusing that Table 1 refers to both activity status and exercise, Table 2 refers to physical activity in the table heading and to ADL in the table, Table 3 refers to PA in the table and to ADL in the note explaining the acronyms in the table. Use the same name consistently for what is the same thing.

260: physical activity = ADL or hours of weekly exercise?

A: As recommended, we have revised “physical activity” to “activity of daily life (ADL)” refer to the same item in Table 2, Table 3, and our manuscript.

In addition, there are different concepts about “activity status”, “exercise” and “activity of daily life” in Table 1and Table 2.

260: 'Significantly negative' : Change to 'significantly and negatively'

A: As recommendation, we have changed “Significantly negative” to “significantly and negatively” (Line 285).

261: postive > positive

A: As recommendation, we have changed “positive” to “positive” (Line 287).

Reviewer 2 Report

This is a study that investigated the factors behind frailty in the elderly to meet the needs of aging, but the poor analysis of previous studies does not effectively convey the value of this study to the reader. 

The author describes SET as follows, but "self-perception of being old" is not found anywhere in Figure 1. Age will be the true age, not "self-perception". 

57-58: An individual’s self-perception of being old will be triggered and fulfilled by some behaviors and physiological responses when this individual ages to a point time [6]. 

The author states as follows, but there is almost no discussion about SET, including the previous related research and their limitation. Furthermore, the discussion part does not mention SET at all. The deficits of previous studies and how the current study compensated them should be explicitly stated. 

95-96: Figure 1 presents the conceptual model developed and tested in this study, which was based on the stereotype embodiment theory (SET) developed by Levy (2009). 

The meaning of the following sentences is difficult to understand. Why do face-oriented Chinese ask a general doctor instead of a psychiatrist for a diagnosis when they are depressed? The author needs to add words to explain. 

312-315: Under the traditional Chinese culture with more concerns of “face”, “endurance” and “stigma”, Chinese older adults with depression demonstrate more somatic complaints [42] leading to their first seeking treatment from general medical setting instead of professional psychiatric assistance and poorer interpersonal relationship with a reduction a social support [41].

The results indicate that social support does not correlate with cognitive function or frailty. Since the current study is an empirical study, it shouldn't recommend to readers anything that has not been gained from the study. The recommendation in this part should be limited to the findings obtained from the analysis of this paper. The argument that social support lowers depression and frailty does not hold because a directional arrow is drawn from depression to social support. If you want to say this, you need to recreate your hypothesis. 

324-328: Further, good interpersonal relationships with strong social support should be established for improving depressive symptoms [47] with subsequent reduction of frailty. 
In sum, it is recommended that older adults should keep regular daily activities, exercise and more social interaction to improve their cognitive function and slow down the processes of frailty.

What is written in the conclusion section is the limitation, not the conclusion. The important claims of this paper should be written in the conclusion section 

Author Response

  1. The author describes SET as follows, but "self-perception of being old" is not found anywhere in Figure 1. Age will be the true age, not "self-perception". 

57-58: An individual’s self-perception of being old will be triggered and fulfilled by some behaviors and physiological responses when this individual ages to a point time [6]. The author states as follows, but there is almost no discussion about SET, including the previous related research and their limitation. Furthermore, the discussion part does not mention SET at all. The deficits of previous studies and how the current study compensated them should be explicitly stated. 95-96: Figure 1 presents the conceptual model developed and tested in this study, which was based on the stereotype embodiment theory (SET) developed by Levy (2009). 

A: 1-1. Thank you for your recommendation and reminding!

In this study, we used the stereotype embodiment theory, we focus on the three tracks of this theory, including psychological, behavioral, and physiological. Aging probably is negatively associated with cognitive function and social support and is positively associated with depression, all these age-related factors can directly and/or indirectly influence their physical health (frailty).

1-2. We did not measure participants' "self-perception of being old", because it's not the purpose of this study, but indeed it's a limitation of this study, and we have added it as a limitation in the manuscript.

1-3. In addition, we have revised our framework to examine the association among cognitive function, depression, social support, and frailty (Figute1). 

2). The meaning of the following sentences is difficult to understand. Why do face-oriented Chinese ask a general doctor instead of a psychiatrist for a diagnosis when they are depressed? The author needs to add words to explain. 312-315: Under the traditional Chinese culture with more concerns of “face”, “endurance” and “stigma”, Chinese older adults with depression demonstrate more somatic complaints [42] leading to their first seeking treatment from general medical setting instead of professional psychiatric assistance and poorer interpersonal relationship with a reduction a social support 

A: We have added some sentences to explainface-oriented Chinese ask a general doctor instead of a psychiatrist for a diagnosis” in this part (Lines 348-355).

3-1). The results indicate that social support does not correlate with cognitive function or frailty. Since the current study is an empirical study, it shouldn't recommend to readers anything that has not been gained from the study. The recommendation in this part should be limited to the findings obtained from the analysis of this paper. The argument that social support lowers depression and frailty does not hold because a directional arrow is drawn from depression to social support. If you want to say this, you need to recreate your hypothesis. 

3-2). 324-328: Further, good interpersonal relationships with strong social support should be established for improving depressive symptoms [47] with subsequent reduction of frailty. In sum, it is recommended that older adults should keep regular daily activities, exercise and more social interaction to improve their cognitive function and slow down the processes of frailty. What is written in the conclusion section is the limitation, not the conclusion. 

The important claims of this paper should be written in the conclusion section 

A: 

3-1) A: As recommendation, we have revised our hypothesis that

a directional arrow is drawn from social support to depression (Lines 348-355 and Figure 2).

3-2)A: As a recommendation, we have added the conclusion section in the manuscript (Lines 376-386).

Round 2

Reviewer 2 Report

The authors have amended appropriately refering my comments.